# Socio-economic factors associated with adolescent pregnancy and motherhood: Analysis of the 2017 Ghana maternal health survey

Ephraim Kumi Senkyire[1]*, Dennis Boateng[2], Felix Oppong Boakye[2], Divine Darlington Logo[3], Magdalena Ohaja[4]

1 Ga West Municipal Hospital, Ghana Health Service, Amasaman-Accra, Ghana, 2 Global Statistical Consult, Accra, Ghana, 3 Research and Development Division, Ghana Health Service, Accra, Ghana, 4 University of Galway, Galway, Ireland

* senkyire88@gmail.com

## Abstract

### Background

Adolescent pregnancy and motherhood have been linked to several factors stemming from social, cultural and to a large extent economic issues. This study examined the socio-economic factors associated with adolescent pregnancy and motherhood in Ghana.

### Design

This was a secondary analysis of the 2017 Ghana Maternal Health Survey, which was a nationally representative cross-sectional survey. Data from 4785 adolescents aged between 15–19 years were included in the analysis. Adolescent pregnancy was defined as adolescents who have ever been pregnant, whiles adolescent motherhood was defined as adolescents who have ever given birth. Weighted logistic regression was used to assess the association between the socio-economic variables and adolescent pregnancy and motherhood.

### Results

Of the 25062 women aged between 15 and 49 years included in the 2017 maternal health survey, 4785 (19.1%) were adolescents between 15–19 years. Adolescent pregnancy was reported in 14.6% (CI:13.2% -16.1%) of the respondents, whereas 11.8% (CI: 10.5% -13.1%) of the respondents had ever given birth. In the multivariate regression analysis, zone (p<0.001), wealth index (p<0.001), age (p<0.001), marital status (p<0.001) and level of education (p<0.001) were all significantly associated with adolescent pregnancy and motherhood. The odds of pregnancy and motherhood were significantly higher in the Middle and Coastal zones (p<0.001), and among older adolescents (p<0.001). However, the odds of pregnancy and motherhood was significantly lower among adolescents from households

**Data Availability Statement:** Complete data are not publicly available but can be requested from the DHS program on reasonable request. The IRB-approval procedures for DHS public-use datasets

do not allow in anyway the respondents, households or sample communities to be identified. To have access to the data, a registered request of a research project must be submitted and approved with the DHS. The instruction for requesting for the Demographic and Health Survey (DHS) data can be found on their website (https://dhsprogram.com/data/Access-Instructions.cfm). The DHS Program will normally review all data requests within 24 – 48 hours (Monday - Friday), and provide notification if access has been granted or additional project information is needed, before access can be granted. The authors confirm that others would be able to access these data in the same manner as themselves. The authors also confirm that they did not have any special access privileges.

**Funding:** The author(s) received no specific funding for this work

**Competing interests:** The authors have declared that no competing interests exist.

with the highest wealth index (p<0.001), among those who were never married (p<0.001) and among adolescents who had secondary/higher education (p<0.001).

## Conclusion

Several socio-economic variables including education, household wealth, marital status and zone of residence were significantly associated with adolescent pregnancy and adolescent motherhood. Sexual and reproductive health education should be intensified among these populations. Adolescent friendly corners should be made available and accessible to all adolescents in Ghana irrespective of where they live or their age.

## Introduction

The United Nations Children's Fund (UNICEF) defines adolescent pregnancy as "an adolescent girl, usually between the ages of 13 and 19 becoming pregnant" [1]. Adolescent pregnancy is a global menace that occurs in both high income and Low- and Middle-Income Countries (LMICs) [2, 3]. However, it is more prevalent in poorly privileged communities [2]. Approximately 21 million girls aged 15–19 years become pregnant annually, and more than half of these girls give birth. It is also worth noting that approximately 777,000 of these births are among adolescent girls below 15 years of age living in LMICs [2].

Adolescent pregnancy is a known contributing factor to the global maternal mortality rate owing to the high incidence of unsafe abortion practices among these age groups [2, 4]. Adolescent motherhood is a vital concern in maternal and child health [5]. The dearth of care among adolescent mothers has advanced to a surged peril of poor maternal and neonatal health sequelae [6]. This predisposes adolescent mothers to a greater risk of eclampsia, prolonged labour, puerperal endometritis, STIs and systemic infections [2, 4, 7, 8]. Consequently, the infants of adolescent mothers face greater risks of low birth weight, preterm delivery and severe neonatal conditions [2, 4, 5, 7, 8]. Evidence exists that children born to adolescent mothers are likely to become adolescent mothers in the future [5].

Adolescent pregnancy and motherhood have been linked to social, cultural and economic factors that affect sexual and reproductive experiences [7]. The social sequelae of adolescent motherhood include isolation by parents and friends, stigma, poverty, unemployment, school disruption and intimate partner violence [2, 5, 6, 9, 51].

The adolescent-specific fertility rate has reduced by 11.6% over the last two decades with large variations across countries: approximately 2% in China to about 18% in Latin America and the Caribbean, and more than 50% in Sub-Saharan Africa [2]. Nevertheless, in LMICs, adolescent birth is still on the rise [2].

The narrative is not different in Ghana where among all births registered in 2014, 30% were from adolescent mothers, with the highest prevalence in the rural setting [3, 10, 15]. Furthermore, in 2017 alone, close to 14% of adolescents aged between 15 and 19 years had already started childbearing [11]. Collectively, adolescent pregnancy contributes to about 9% of maternal mortality in Ghana [12]. It is important to note that relatively little research has been conducted on the socio-economic consequences of adolescent pregnancy and motherhood in Ghana. Although there are several studies on adolescent pregnancy in Ghana, few studies have examined the association of socio-economic factors on adolescent pregnancy and motherhood using data from nationally representative surveys [13, 14].

These studies mostly used data from the past Ghana demographic and health surveys with a focus on adolescent fertility rates. Therefore, this study sought to assess the association of socio-economic factors with adolescent pregnancy and motherhood in Ghana. Understanding both the social and economic influences of these variables are vital for effective policy formulations [15].

## Methods

The data used for this study were obtained from the 2017 Ghana Maternal Health Survey (GHMS) [11] which was implemented by the Ghana Statistical Service. The data collected in the survey include individual and household level data. The design and methods used make it possible to obtain representative estimates across the whole country for maternal mortality. The sampling frame utilised in the 2017 GMHS was established from the 2010 Population and Housing Census (PHC) in Ghana [11]. The survey sampling technique consisted of a two-stage stratification procedure. In the interstratification stage, each of the 10 regions of Ghana was separated into rural and urban areas to generate a total of 20 sampling strata.

An independent selection in each stratum occurred in two stages, first with the sorted strata generated from administrative regions and levels using implicit stratification and proportional allocation before sample selection. Initially, a proportional probability sampling technique was used to select a total of 900 enumeration areas consisting of all regions. A cluster size of 466 was produced from urban areas and 434 from rural areas. In the second stage, 30 households were randomly sampled from each of the 900 clusters to produce a total sample size of 27000 households. From these households 20277 (80.9%) women were aged 20 years and above whiles, 4785 (19.1%) were adolescents aged between 15–19 years. Of 4785 adolescents aged between 15–19 years, 701 (14.6%) had ever been pregnant and 566 (11.8%) had ever had a live birth. Further details on the data description are presented in Fig 1. Only adolescents aged 15–19 years were included in this secondary data analysis.

### Study variables

A detailed description of the variables included in this secondary analysis are listed in Table 1. The variable definitions and how they were utilised in the data analysis are presented.

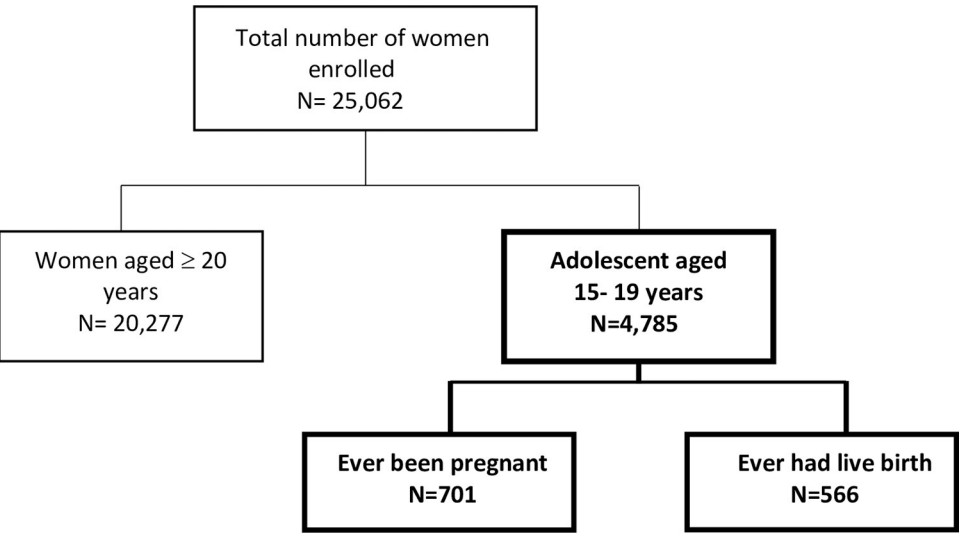

**Fig 1. Description of study respondents.** Respondents included in this secondary analysis are highlighted.

**Table 1. Definitions of study variables and their utilisation in models formulated.**

| Variable | Definition | Utilised in Model/Type |
|---|---|---|
| Adolescent motherhood | Adolescents who have ever given birth. | Dependent/Binary |
| Adolescent pregnancy | Adolescents who have ever been pregnant. | Dependent/Binary |
| Place of residence | Location of household (Urban; Rural) | Independent/Binary |
| Region of residence | Formulated regional locations with 10 administrative regions (Western; Central; Greater Accra; Volta; Eastern; Ashanti; Brong Ahafo; Northern; Upper East; Upper West) | Independent/Categorical |
| Zone | The location from three major geographic regions in Ghana, i.e. **Coastal**: Western, Central, Greater Accra, and Volta regions. **Middle**: Eastern, Ashanti, and Brong Ahafo regions **Northern**: Northern, Upper East, and Upper West regions. | Independent/Categorical |
| Wealth index | The wealth status of households is grouped as Lowest, Second, Middle, Fourth and Highest. | Independent/Categorical |
| Age | Age of women grouped into <15 years, 16 years, 17 years, 18 years, and 19 years. | Independent/Categorical |
| Marital status | Grouped as Never Married; Married; Living together with a man; Divorced/Separated. | Independent/Categorical |
| Media exposure | Women's access to media including the internet, print, television and radio at least once a week are grouped as Yes; No. | Binary |
| Level of education | Women's highest level of education (No education; Primary education; Middle/Junior Secondary/High School (JSS/JHS); Secondary /Higher education) | Independent/Categorical |

## Statistical analysis

Data analysis was performed using SAS (Statistical Analysis Software 9.4, SAS Institute Inc, Cary, North Carolina, USA). Sampling weights were used to obtain a national/regional representation of the survey results. Weights were calculated separately for each sampling stage and each cluster using probability sampling. The proportion of women aged 15–19 years who have ever been pregnant (adolescent pregnancy) and those who have ever given birth (adolescent motherhood) were presented by the different demographic variables (place/region/zone of residence, wealth index, age of woman, marital status, media exposure and level of education of woman). Chi-square tests were used to assess the association of demographic variables with adolescent pregnancy and adolescent motherhood.

The dependent variables, adolescent pregnancy and adolescent motherhood were defined for women aged 15–19 years. Independent variables included in the analysis were a place of residence, zone, wealth index, age, marital status, media exposure and educational level. Both dependent variables were coded as binary variables and fitted in a weighted logistic regression analysis. Univariate and multivariate techniques were used to assess the association of socio-economic variables with adolescent pregnancy and adolescent motherhood.

All variables were included in both univariate and multivariate analysis except for the region of residence that was excluded due to the many levels and aggregated into the zone for ease and clarity of interpretation. Also, given the interest in the a priori selected variables, they were all included in the adjusted analysis without considering their level of significance in the unadjusted analysis. Unadjusted/adjusted odds ratios with 95% confidence intervals were computed, and variables with p-value <0.05 in the univariate/multivariate analyses were considered statistically significant.

### Ethical consideration

The study protocol was reviewed and approved by the ICF institutional review board [11]. Informed written and signed consent was provided by all study respondents for their participation in the survey. The structured data collection tool administered by trained data enumerators was translated where necessary from the English language to a local dialect to obtain responses. Further details on the survey design and methodology can be found in the survey report [11].

## Results

### Characteristics of study participants

Out of the 25062 women included in the 2017 maternal health survey, 4785 were adolescents aged 15–19 years. The mean age of the 4758 respondents included in this analysis was 17 (Standard Deviation = 1.40) years. More than half (54.2%) of the study participants had middle/JSS/JHS education. Compared to the other regions, most of the study participants were from the Ashanti region (19.6%). A significant majority (90.9%) of the respondents were never married, with (0.6%) being divorced or separated. A similar percentage (91.0%) of the adolescent respondents were exposed to at least one form of media. See Table 2.

### Characteristics of study participants by pregnancy and birth history

At the time of the survey, 123/4785 respondents (2.6%, 95% CI: 2.2 - 2.9%) were pregnant. As presented in Table 3, 14.6% (95% CI: 13.2 - 16.1%) of the study respondents had ever been pregnant, whereas 11.8% (95% CI: 10.5 - 13.1%) had ever given birth. Adolescent pregnancy was higher in rural areas compared to urban areas (17.6% vs. 11.7%) and differed by region of residence (p<0.001), with the highest prevalence in the Brong Ahafo Region (18.6% 95% CI:14.9% - 22.4%) and the lowest prevalence in the Greater Accra region (8.1% 95% CI: 6.1% - 10.2%). Respondents from households with the highest wealth index had the lowest prevalence. Adolescent pregnancy increased with age, with prevalence lower among younger adolescents and higher among older adolescents, whereas those from a household with the lowest and second-lowest wealth index had a high prevalence of adolescent pregnancy (p<0.001). Similarly, adolescent pregnancy was significantly lower among respondents with secondary/higher education (p<0.001).

Similar results were reported for the respondents who have had a live birth–higher in rural areas, higher among respondents from a household with the lowest and second-lowest wealth index, higher among older adolescents and higher among adolescents with no education. Also, adolescent motherhood was higher among older adolescents than younger adolescents (p<0.001).

### Association of socio-economic factors with adolescent pregnancy

In univariate analysis, place of residence (p<0.001), wealth index (p<0.001), age (p<0.001), marital status (p<0.001), mass media exposure (p = 0.022) and level of education (p<0.001) were significantly associated with adolescent pregnancy. In multivariate analysis, zone (p<0.001), wealth index (p<0.001), age (p<0.001), marital status (p<0.001) and level of education (p<0.001) were significantly associated with adolescent pregnancy. The odds of pregnancy were significantly higher in the middle zone (AOR = 2.96, 95% CI: 1.95–4.52) and coastal zone (AOR = 3.71, 95% CI: 2.37–5.80) compared to the northern zone. With regards to the wealth index, the odds of pregnancy were significantly lower among adolescents from households with the highest wealth index compared to adolescents from households with the

**Table 2. Characteristics of study participants (adolescents aged 15–19 years).**

| Variable | Unweighted frequency | Weighted frequency | Weighted percentage |
|---|---|---|---|
| **Place of residence** | | | |
| Urban | 2240 | 2411 | 50.4 |
| Rural | 2648 | 2374 | 49.6 |
| **Region of residence** | | | |
| Western | 463 | 650 | 13.6 |
| Central | 290 | 413 | 8.6 |
| Greater Accra | 405 | 748 | 15.6 |
| Volta | 286 | 396 | 8.3 |
| Eastern | 421 | 490 | 10.2 |
| Ashanti | 606 | 936 | 19.6 |
| Brong Ahafo | 479 | 490 | 10.2 |
| Northern | 817 | 351 | 7.3 |
| Upper east | 556 | 176 | 3.7 |
| Upper west | 565 | 136 | 2.8 |
| **Zone** | | | |
| Northern | 1938 | 663 | 13.9 |
| Middle | 1506 | 1916 | 40.0 |
| Coastal | 1444 | 2206 | 46.1 |
| **Wealth Index** | | | |
| Lowest | 1503 | 869 | 18.2 |
| Second | 1011 | 1027 | 21.5 |
| Middle | 878 | 1051 | 22.0 |
| Fourth | 823 | 953 | 20.0 |
| Highest | 673 | 886 | 18.5 |
| **Age** | | | |
| 15 years | 1203 | 1046 | 21.9 |
| 16 years | 927 | 936 | 19.6 |
| 17 years | 1047 | 1098 | 23.0 |
| 18 years | 959 | 974 | 20.4 |
| 19 years | 752 | 731 | 15.3 |
| **Marital Status** | | | |
| Never Married | 4403 | 4350 | 90.9 |
| Married | 176 | 87 | 1.8 |
| Living together with a man | 279 | 317 | 6.6 |
| Divorced/separated | 30 | 31 | 0.6 |
| **Media exposure (Internet/print/television/radio)** | | | |
| Yes | 4183 | 4354 | 91.0 |
| No | 705 | 430 | 9.0 |
| **Level of education** | | | |
| No education | 271 | 164 | 3.4 |
| Primary | 948 | 835 | 17.5 |
| Middle/JSS/JHS | 2609 | 2595 | 54.2 |
| Secondary/ Higher | 1060 | 1191 | 24.8 |

lowest wealth index (AOR = 0.28, 95% 0.14–0.55). The odds of pregnancy increased with age: AOR = 1.92, 5.08, 10.50 and 20.27 for adolescents aged 16 years, 17 years, 18 years and 19 years respectively compared to those aged 15 years. Married adolescents (AOR = 16.48, 95%

**Table 3. Background characteristics of adolescents aged 15–19 years by pregnancy and birth history.**

| Variable | Total number of women aged 15–19 years | Adolescents aged 15–19 years who have ever been pregnant | | | Adolescents aged 15–19 years who have had a live birth | | |
|---|---|---|---|---|---|---|---|
| | N | N | % (95% CI) | p-value | N | % (95% CI) | p-value |
| Overall | 4785 | 701 | 14.6 (13.2–16.1) | - | 566 | 11.8 (10.5–13.1) | - |
| **Place of residence** | | | | | | | |
| Urban | 2411 | 283 | 11.7 (9.8–13.7) | 0.001 | 222 | 9.2 (7.5–11.0) | 0.009 |
| Rural | 2374 | 418 | 17.6 (15.1–20.1) | | 344 | 14.5 (12.1–16.9) | |
| **Region of residence** | | | | | | | |
| Western | 650 | 120 | 18.5 (12.5–24.3) | <0.001 | 102 | 15.7 (10.3–21.0) | <0.001 |
| Central | 413 | 71 | 17.2 (10.0–24.5) | | 56 | 13.6 (6.8–20.3) | |
| Greater Accra | 748 | 61 | 8.1 (6.1–10.2) | | 51 | 6.8 (4.9–8.6) | |
| Volta | 396 | 61 | 15.4 (7.3–23.5) | | 54 | 13.6 (5.8–21.7) | |
| Eastern | 490 | 67 | 13.7 (10.6–16.5) | | 54 | 11.0 (8.6–13.7) | |
| Ashanti | 936 | 147 | 15.7 (11.5–20.0) | | 114 | 12.2 (8.4–16.0) | |
| Brong Ahafo | 490 | 91 | 18.6 (14.9–22.4) | | 65 | 13.3 (10.0–16.5) | |
| Northern | 351 | 48 | 13.7 (11.4–16.2) | | 41 | 11.7 (9.4–13.7) | |
| Upper east | 176 | 26 | 14.8 (11.4–17.6) | | 22 | 12.5 (10.2–15.3) | |
| Upper west | 136 | 10 | 7.4 (5.1–8.8) | | 6 | 4.4 (2.9–6.6) | |
| **Zone** | | | | | | | |
| Northern | 663 | 84 | 10.6 (9.0–12.0) | <0.001 | 70 | 10.6 (9.0–12.0) | <0.001 |
| Middle | 1916 | 305 | 12.2 (10.1–14.4) | | 234 | 12.2 (10.1–14.4) | |
| Coastal | 2206 | 312 | 12.0 (9.7–14.1) | | 263 | 12.0 (9.7–14.1) | |
| **Wealth Index** | | | | | | | |
| Lowest | 869 | 167 | 19.2 (14.7–24.1) | <0.001 | 146 | 16.8 (12.4–21.2) | <0.001 |
| Second | 1027 | 218 | 21.2 (17.0–25.5) | | 190 | 18.5 (14.5–22.4) | |
| Middle | 1051 | 180 | 17.1 (13.4–20.8) | | 139 | 13.2 (9.8–16.6) | |
| Fourth | 953 | 103 | 10.8 (8.1–13.5) | | 70 | 7.3 (5.2–9.5) | |
| Highest | 886 | 33 | 3.7 (2.1–5.2) | | 22 | 2.5 (1.2–3.7) | |
| **Age** | | | | | | | |
| 15 years | 1046 | 34 | 3.2 (1.8–4.7) | <0.001 | 29 | 2.8 (1.4–4.0) | <0.001 |
| 16 years | 936 | 50 | 5.3 (3.6–7.1) | | 40 | 4.3 (2.7–5.8) | |
| 17 years | 1098 | 144 | 13.1 (10.1–16.1) | | 107 | 9.7 (7.0–12.4) | |
| 18 years | 974 | 225 | 23.1 (19.0–27.2) | | 173 | 17.8 (14.4–21.0) | |
| 19 years | 731 | 248 | 33.9 (28.6–39.3) | | 218 | 29.8 (24.6–35.0) | |
| **Marital Status** | | | | | | | |
| Never Married | 4350 | 376 | 8.6 (7.5–9.7) | <0.001 | 273 | 6.3 (5.2–7.3) | <0.001 |
| Married | 87 | 59 | 67.8 (49.4–49.4) | | 52 | 59.7 (42.5–77.1) | |
| Living together with a man | 317 | 241 | 76.0 (59.3–92.7) | | 216 | 68.1 (52.7–83.9) | |
| Divorced/separated | 31 | 26 | 84.0 (41.9–122.6) | | 25 | 80.6 (38.7–119.4) | |
| **Media exposure (Internet/print/television/radio)** | | | | | | | |
| Yes | 4354 | 620 | 14.2 (12.8–15.7) | <0.001 | 499 | 11.5 (10.1–12.8) | <0.001 |
| No | 430 | 82 | 19.1 (19.1–24.2) | | 67 | 15.6 (10.7–20.2) | |
| **Level of education** | | | | | | | |
| No education | 164 | 51 | 31.1 (21.3–40.9) | <0.001 | 45 | 27.4 (17.7–36.6) | <0.001 |
| Primary | 835 | 191 | 22.9 (18.6–27.7) | | 166 | 19.9 (15.4–24.3) | |
| Middle/JSS/JHS | 2595 | 391 | 15.1 (13.3–16.8) | | 319 | 12.3 (10.6–14.0) | |
| Secondary/Higher | 1191 | 68 | 5.9 (4.1–7.7) | | 36 | 3.0 (1.7–4.5) | |

**Table 4. Association of demographic factors with adolescents (aged 15–19 years) adolescent pregnancy.**

| Variable | Univariate analysis | | | Multivariate analysis | | |
|---|---|---|---|---|---|---|
| | Odd ratio | 95% CI | p-value | Adjusted Odd ratio | 95% CI | p-value |
| **Place of residence** | | | | | | |
| Rural | 1 | | <0.001 | 1 | | 0.884 |
| Urban | 0.62 | 0.48–0.81 | | 1.03 | 0.74–1.43 | |
| **Zone** | | | | | | |
| Northern | 1 | | 0.093 | 1 | | <0.001 |
| Middle | 1.32 | 1.03–1.68 | | 2.96 | 1.95–4.52 | |
| Coastal | 1.14 | 0.88–1.48 | | 3.71 | 2.37–5.80 | |
| **Wealth Index** | | | | | | |
| Lowest | 1 | | <0.001 | 1 | | <0.001 |
| Second | 1.13 | 0.82–1.56 | | 1.03 | 0.68–1.56 | |
| Middle | 0.87 | 0.62–1.21 | | 0.93 | 0.55–1.59 | |
| Fourth | 0.51 | 0.36–0.73 | | 0.76 | 0.45–1.27 | |
| Highest | 0.16 | 0.10–0.26 | | 0.28 | 0.14–0.55 | |
| **Age** | | | | | | |
| 15 years | 1 | | <0.001 | 1 | | <0.001 |
| 16 years | 1.68 | 0.98–2.88 | | 1.92 | 1.12–3.30 | |
| 17 years | 4.48 | 2.72–7.40 | | 5.08 | 3.07–8.42 | |
| 18 years | 8.90 | 5.53–14.32 | | 10.50 | 6.34–17.39 | |
| 19 years | 15.24 | 9.47–24.52 | | 20.27 | 11.94–34.42 | |
| **Marital status** | | | | | | |
| Never Married | 1 | | <0.001 | 1 | | <0.001 |
| Married | 21.90 | 13.84–34.63 | | 16.48 | 7.96–34.10 | |
| Living together with a man | 33.51 | 23.01–48.79 | | 17.24 | 11.80–25.19 | |
| Divorced/separated | 52.05 | 17.41–155.65 | | 14.30 | 4.62–44.27 | |
| **Media exposure(Internet/print/television/radio)** | | | | | | |
| Yes | 1 | | 0.022 | 1 | | 0.632 |
| No | 1.41 | 1.05–1.90 | | 1.11 | 0.74–1.66 | |
| **Level of education** | | | | | | |
| No education | 1 | | <0.001 | 1 | | <0.001 |
| Primary | 0.67 | 0.44–1.00 | | 2.04 | 1.02–4.06 | |
| Middle/JSS/JHS | 0.40 | 0.27–0.60 | | 1.01 | 0.51–1.99 | |
| Secondary/Higher | 0.14 | 0.08–0.23 | | 0.29 | 0.13–0.64 | |

CI:7.96–34.10), those living together with a man (AOR = 17.24, 95% CI:11.80–25.19) and those divorced/separated (AOR = 14.30, 95% CI: 4.62–44.27) had significantly higher odds of pregnancy compared those who were never married. Adolescents who had secondary/higher education had significantly lower odds of pregnancy compared to those with no formal education (AOR = 0.29, 95% CI: 0.13–0.64). See Table 4.

## Association of socio-economic factors with adolescent motherhood

In univariate analysis, place of residence (p<0.001), wealth index (p<0.001), age (p<0.001), marital status (p<0.001), mass media exposure (p = 0.013) and level of education (p<0.001) were significantly associated with adolescent motherhood. In multivariate analysis, zone (p<0.001), wealth index (p<0.001), age (p<0.001), marital status (p<0.001) and level of education (p<0.001) were significantly associated with adolescent motherhood.

The odds of adolescent motherhood were significantly higher in the middle zone (AOR = 2.79, 95% CI: 1.75–4.46) and coastal zone (AOR = 4.44, 95% CI: 2.71–7.26) compared to the Northern zone. With regards to the wealth index, the odds of motherhood were significantly lower among adolescents from households with the highest wealth index compared to adolescents from households with the lowest wealth index (AOR = 0.23, 95% 0.11–0.47). The odds of motherhood increased with age: AOR = 1.78, 4.01, 8.07 and 19.50 for adolescents aged 16 years, 17 years, 18 years and 19 years respectively compared to those aged 15 years. Married adolescents (AOR = 16.48, 95% CI:7.63–34.55), those living together with a man (AOR = 17.24, 95% CI:11.15–23.85) and those divorced/separated (AOR = 14.30, 95% CI: 6.48–53.51) had significantly higher odds of motherhood compared those who were never married. Adolescents who had secondary/higher education had significantly lower odds of motherhood compared to those with no formal education (AOR = 0.21, 95% CI: 0.09–0.50). See Table 5.

**Table 5. Association of demographic factors with adolescents (aged 15–19 years) motherhood.**

| Variable | Univariate analysis | | | Multivariate analysis | | |
|---|---|---|---|---|---|---|
| | Odd ratio | 95% CI | p-value | Adjusted Odd ratio | 95% CI | p-value |
| **Place of residence** | | | | | | |
| Rural | 1 | | <0.001 | 1 | | 0.919 |
| Urban | 0.60 | 0.45–0.80 | | 0.98 | 0.68–1.42 | |
| **Zone** | | | | | | |
| Northern | 1 | | 0.486 | 1 | | <0.001 |
| Middle | 1.19 | 0.89–1.58 | | 2.79 | 1.75–4.46 | |
| Coastal | 1.16 | 0.87–1.54 | | 4.44 | 2.71–7.26 | |
| **Wealth Index** | | | | | | |
| Lowest | 1 | | <0.001 | 1 | | <0.001 |
| Second | 1.13 | 0.80–1.58 | | 0.97 | 0.63–1.50 | |
| Middle | 0.75 | 0.52–1.10 | | 0.72 | 0.40–1.30 | |
| Fourth | 0.40 | 0.27–0.59 | | 0.54 | 0.30–0.96 | |
| Highest | 0.13 | 0.07–0.23 | | 0.23 | 0.11–0.47 | |
| **Age** | | | | | | |
| 15 years | 1 | | <0.001 | 1 | | <0.001 |
| 16 years | 1.59 | 0.85–2.97 | | 1.78 | 0.96–3.32 | |
| 17 years | 3.85 | 2.18–6.80 | | 4.01 | 2.25–7.14 | |
| 18 years | 7.68 | 4.57–12.90 | | 8.07 | 4.62–14.12 | |
| 19 years | 15.18 | 8.94–25.78 | | 19.50 | 10.67–35.64 | |
| **Marital status** | | | | | | |
| Never Married | 1 | | <0.001 | 1 | | <0.001 |
| Married | 22.19 | 13.98–35.23 | | 16.24 | 7.63–34.55 | |
| Living together with a man | 32.34 | 22.47–46.56 | | 16.31 | 11.15–23.85 | |
| Divorced/separated | 61.05 | 21.97–169.64 | | 18.63 | 6.48–53.51 | |
| **Media exposure(Internet/print/television/radio)** | | | | | | |
| Yes | 1 | | 0.013 | 1 | | 0.878 |
| No | 1.42 | 1.03–1.95 | | 0.97 | 0.62–1.50 | |
| **Level of education** | | | | | | |
| No education | 1 | | <0.001 | 1 | | <0.001 |
| Primary | 0.70 | 0.44–1.03 | | 2.05 | 1.03–4.11 | |
| Middle/JSS/JHS | 0.38 | 0.25–0.58 | | 1.02 | 0.52–2.02 | |
| Secondary/Higher | 0.08 | 0.05–0.16 | | 0.21 | 0.09–0.50 | |

## Discussion

Adolescent pregnancy and motherhood are major challenges facing low resourced countries in Sub-Saharan Africa (SSA) including Ghana [3]. Pregnant adolescents and mothers face diverse challenges including physical, psychological, mental and social obstacles and pregnancy-related challenges [16–30]. This paper explored the association of socioeconomic factors with adolescent pregnancy and motherhood. The prevalence of adolescent pregnancy was 14.6%, while the prevalence of adolescent motherhood was 11.8%. The former was inconsequential compared to 14% in 2014 [31]. Adolescent pregnancy is lower in Ghana compared to other African countries. A prevalence of 18% has been reported in Kenya [32], 19% in Nigeria [33] and 36% in Mali [34]. Again the prevalence of AP in Ghana is lower compared to the overall 18.8% prevalence in Africa and the 19.3% prevalence in SSA [35]. The disparities could be a consequence of the existence of several cultural, sociodemographic, and individual adolescent features.

Significant regional variations were found in the prevalence of adolescent pregnancy and adolescent motherhood in Ghana, with a high AP and AM prevalence in the Brong-Ahafo, Western and Central regions. Likewise, the 2014 GDHS indicated that adolescent girls residing in the Brong Ahafo, Central and Volta regions start childbearing earlier than adolescents in other regions [35]. These differences have been attributed to poverty and employment satus [36], transactional sex [30, 37, 38], decline in menarche [39, 40], child marriage [41], early sexual debut [42–45], lack of contraceptive knowledge [46, 47] and inadequate sexual and reproductive health education [48].

From our analysis, it is evident that socio-economic factors are significantly associated with adolescent pregnancy and adolescent motherhood. Several studies in Ghana have also reported the association of socio-economic factors with adolescent pregnancy and adolescent motherhood [6, 14, 48–55]. In Asare et al. [56], adolescents from low economic backgrounds were about 4 times more likely to be pregnant compared to those from high-income households. This was consistent with the results of our study where adolescent pregnancy and adolescent motherhood was found to be significantly higher among adolescents from low-income households. Other studies in Sub-Saharan Africa associated the high prevalence of adolescent pregnancies with low socio-economic status [57–59].

Further, AP and AM were noted to be associated with increasing age, with prevalence lower among younger adolescents and higher among older adolescents. This was also revealed in studies done by Uwizeye et al. [60] and Habitu et al. [61] where age was significantly associated with AP and AM and age at first sexual debut increased AP and AM [62, 63]. This could be attributed to older adolescents having access to "virulent" digital information [60], peer influence and increasing sexual drive as age increases [61].

Although in the multivariate analysis, there was no significant association between place of residence and media and adolescent pregnancy/adolescent motherhood, other studies reported that residence [43, 61, 64, 65] and media [53, 66] were associated with adolescent pregnancy/adolescent motherhood. In contrast, a study in Ethiopia indicated that early sexual debut is more prevalent among urban residents [67]. Likewise, adolescents in rural West Africa had lower odds of first pregnancy [68].

In multivariate analysis, zone and marital status were significantly associated with adolescent pregnancy and motherhood. The odds of pregnancy and motherhood were significantly higher in the middle zone and coastal zone compared to the northern zone. This was similar in the 2014 GDHS report where the Brong Ahafo, Central, and Volta regions were among the regions with the highest adolescent pregnancy [31]. Conversely, shreds of evidence showed that adolescent pregnancy is higher in the northern region due high prevalence of adolescent

marriage [69]. Nevertheless, in the Ghanaian culture, marriage is associated with childbirth because extra-marital sex and early childbearing are scowled and unethical, ergo early marriage is emboldened [70]. Parallelly, the convention by which young women are anticipated to begin child-bearing shortly following marriage is a contributing factor leading adolescent mothers to become pregnant [69]. As found in our study, reports from several studies have shown a significant association between marital status and adolescent pregnancy [61, 68, 71]. The consequences of adolescent marriage are multi-faceted including increased risk for sexually transmitted diseases, cervical cancer, death during childbirth, obstetric fistulas, child mortality and low agency, deprivation of education, violence, abuse and forced sexual relations [69, 72–78].

From both the univariate and multivariate analysis, higher levels of education were associated with reduced odds of adolescent pregnancy and adolescent motherhood. In parallel, Okine and Dako-Gyeke [50], indicated that a low level of education was among the factors contributing to adolescent pregnancy in Ghana. Accordingly, an extra year of schooling was reported to reduce the likelihood of marriage and childbirth before age 18 [79]. Attaining higher education prevents adolescent pregnancy in low-income countries [80]. Huang [81] echoed those girls who have higher education are five times less likely to become pregnant, similar to Mamboreo [64] who reported that level of education influences AM. Adolescents who report a pregnancy in a South African study were less educated [82]. Hence, adolescents with the slightest secondary education had a lower risk of childbirth [61]. Again nulliparous adolescent girls are more likely to receive pregnancy prevention information from school [83]. Yet, mothers were able to complete their education despite being adolescent mothers in another study in South Africa [84]. It is well documented that educated women use maternal care services regularly to prevent both neonatal and maternal mortality. Thus advancing access to basic education among girls is a constructive plan to decrease adolescent pregnancy and adolescent motherhood and their related side effects [8, 79].

## Conclusion

Our analysis revealed that education, age, household wealth, marital status and zone of residence are associated with adolescent pregnancy and adolescent motherhood in Ghana. Given that adolescent pregnancy and motherhood were significantly higher in the Middle and Coastal zones, and among older adolescents, sexual and reproductive health education should be intensified among these populations. Also, adolescent-friendly corners should be made available and accessible to all adolescents in Ghana irrespective of where they live or their age.

## Supporting information

**S1 File.**
(PDF)

## Acknowledgments

The authors are grateful to the DHS program for providing them access to the 2017 Ghana Maternal Health Survey database. We are also grateful to the survey participants.

## Author Contributions

**Conceptualization:** Ephraim Kumi Senkyire.

**Data curation:** Ephraim Kumi Senkyire, Dennis Boateng.

**Formal analysis:** Ephraim Kumi Senkyire.

**Methodology:** Ephraim Kumi Senkyire, Dennis Boateng.

**Project administration:** Ephraim Kumi Senkyire.

**Resources:** Ephraim Kumi Senkyire.

**Software:** Dennis Boateng.

**Supervision:** Ephraim Kumi Senkyire, Felix Oppong Boakye, Magdalena Ohaja.

**Validation:** Ephraim Kumi Senkyire, Felix Oppong Boakye.

**Writing – original draft:** Ephraim Kumi Senkyire.

**Writing – review & editing:** Ephraim Kumi Senkyire, Dennis Boateng, Felix Oppong Boakye, Divine Darlington Logo, Magdalena Ohaja.

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
