## [Decision Letter · Decision Letter 0]

7 Feb 2022

PONE-D-21-40893Socio-economic factors associated with adolescent pregnancy and motherhood: analysis of the 2017 Ghana maternal health surveyPLOS ONE

Dear Corresponding Author,

Thank you for submitting your manuscript to PLOS ONE. After careful consideration, we feel that it has merit but does not fully meet PLOS ONE’s publication criteria as it currently stands. Therefore, we invite you to submit a revised version of the manuscript that addresses the points raised during the review process.

We look forward to receiving your revised manuscript.

Kind regards,

Nguyen Toan Tran

Academic Editor

PLOS ONE

Journal Requirements:

2. Please change "female” or "male" to "woman” or "man" as appropriate, when used as a noun (see for instance https://apastyle.apa.org/style-grammar-guidelines/bias-free-language/gender).

Reviewers' comments:

Reviewer's Responses to Questions

**Comments to the Author**

1. Is the manuscript technically sound, and do the data support the conclusions?

Reviewer #1: Yes

Reviewer #2: Yes

2. Has the statistical analysis been performed appropriately and rigorously? 

Reviewer #1: Yes

Reviewer #2: Yes

3. Have the authors made all data underlying the findings in their manuscript fully available?

Reviewer #1: Yes

Reviewer #2: No

4. Is the manuscript presented in an intelligible fashion and written in standard English?

Reviewer #1: Yes

Reviewer #2: Yes

5. Review Comments to the Author

Reviewer #1: The authors introduced an abbreviation in the abstract; AP and AM. I recommend, the words are written fully. Also, the recommendation is not based on the conclusions of the manuscript, therefore, recommendation should be focused on health promotion tenets such increased education, accessibility to contraceptives at adolescent friendly corners and policies to promote adolescents reproductive health. Again, the authors introduced abbreviation of AP and AM in the introduction but the words still being used in the manuscripts. The authors should make corrections on these mishaps especially in the results session.

Reviewer #2: Socio-economic factors associated with adolescent pregnancy and motherhood: analysis of the 2017 Ghana maternal health survey

Thank you for the opportunity to review this manuscript. I think it has merit for publishing but I have a number of comments that need to be addressed before it can be accepted for publication. These comments are below:

I recommend a critical read to correct grammatical errors/omissions/repetitions/ and omissions. e.g.

a. …was significantly higher in the Middle and Coastal zones (p<0.001), among older adolescents (p<0.001).

b. The design and methods used makes it…

c. … that aff ect sexual…

d. …first with the the sorted…

e. …with the the sorted…

f. kenya

ABSTRACT

Design: The authors should clarify the sample for the study, thus what age frame was the caveat? Motherhood as conceptualised for the study should be defined in this section.

Results: In the first statement, i.e. “Of the 25062 women included in the survey, 4785 were adolescents between 15-19 years.” Does this mean that the study included participants outside adolescence, considering the word “women”. This need to be clarified and if only females in adolescence, then authors should change “women” to “adolescents”. Also, the percentage of 4785 should be captured in the statement.

Conclusion: Write these in full “AP and AM.” From the focus of the study and your findings, it is expected that you make specific recommendation that address your “socio-economic” findings, however, not even a single recommendation was made in this regard. The recommendations provided must therefore be reconsidered.

INTRODUCTION

Page 9: Change from “The Social sequelae…” to “The social sequelae…”

Page 10: Remove the misplaced “s” from “…births are s still on the rise…”

Page 10: Authors should reconstruct this statement for clarity “It is important to note that relatively little research has been conducted on the socio-economic consequences of adolescent mothers and their children in Ghana.”

METHODS

“GMHS” should be written in full at first use.

The last paragraph under the “METHODS” section should be removed and be placed under a subheading “Ethical Consideration”, possibly right after the statistical analysis.

RESULTS

Change from “RESULT” to “RESULTS”

Instead of “adolescents”, some sections of the results mention “women” and this must be revised.

Characteristics of study participants.

Authors should report the standard deviation in addition to the mean age.

Characteristics of study participants by pregnancy and birth history

“…with the highest prevalence in the Brong Ahafo Region and the lowest

prevalence in the Greater Accra region.” Please quote the percentages and CIs.

Association of socio-economic factors with adolescent pregnancy (AP)/ Association of socio-economic factors with adolescent motherhood (AM)

Authors reported some adjusted findings in these sections yet the odds ratios are quoted as though they are results from unadjusted models. This should be rectified for clarity.

DISCUSSION

Great discussion. The findings are well discussed and situated in the body of knowledge.

CONCLUSION

The conclusions are great, however, the recommendations do not align with findings of the study and the prime focus of the study. Recommendations are to be made regarding the socio-economic factors studied.

6. PLOS authors have the option to publish the peer review history of their article (what does this mean?). If published, this will include your full peer review and any attached files.

Reviewer #1: **Yes: **Thomas Hormenu, PhD. University of Cape Coast, Ghana

Reviewer #2: **Yes: **Edward Kwabena Ameyaw

---

## [Author Response · Author response to Decision Letter 0]

14 Apr 2022

Ga West Municipal Hospital

Ghana Health Service, 

Amasaman-Accra, Ghana

2/02/2022

The Managing Editor 

PLOS ONE

Dear Editor, 

ACTIONS AND/OR RESPONSES TO REVIEWER COMMENTS

Thank you so much for giving us an opportunity to submit a revised draft of our manuscript entitled “Socio-economic factors associated with adolescent pregnancy and motherhood: analysis of the 2017 Ghana maternal health survey”. We appreciate the time and effort that you and the reviewers dedicated to providing feedback on our manuscript. We are very grateful for the insightful comments and valuable improvements to our paper. We have incorporated all the suggestions and comments made by the handling editor and the reviewers. We have cautiously revised the article as shown in the detailed point-by-point actions, responses or explanations to all the reviewer comments and suggestions as presented herein. Changes within the main document are highlighted in colour. 

 Reviewer # 1 Responses Line numbers/Page Numbers

1 Please ensure that your manuscript meets PLOS ONE's style requirements, including those for file naming. The PLOS ONE style templates can be found at 

 The current manuscript now meets PLOS ONE style requirements. 

 throughout the manuscript

3 In your Data Availability statement, you have not specified where the minimal data set underlying the results described in your manuscript can be found. PLOS defines a study's minimal data set as the underlying data used to reach the conclusions drawn in the manuscript and any additional data required to replicate the reported study findings in their entirety. All PLOS journals require that the minimal data set be made fully available. For more information about our data policy, please see http://journals.plos.org/plosone/s/data-availability.

 The Data Available Statement has been included in the manuscript 308-330

4 . Please amend either the abstract on the online submission form (via Edit Submission) or the abstract in the manuscript so that they are identical.

 The abstract in both the online submission form and within the manuscript have been amended and are now identical.

 23-47 in the manuscript and updated online

 Reviewer #1: 

1 1. The authors introduced an abbreviation in the abstract; AP and AM. I recommend, the words are written fully. 

 AP and AM have been written in full as adolescent pregnancy and adolescent motherhood respectively.

 Throughout the manuscript 

2 Also, the recommendation is not based on the conclusions of the manuscript, therefore, recommendation should be focused on health promotion tenets such increased education, accessibility to contraceptives at adolescent friendly corners and policies to promote adolescents’ reproductive health. 

 The reviewers’ suggestions have been incorporated into the recommendation. 

 302-306

3 Again, the authors introduced abbreviation of AP and AM in the introduction but the words still being used in the manuscripts. The authors should make corrections on these mishaps especially in the results session. 

 The abbreviations AP and AM are now written in full throughout the manuscript.

 Entire manuscript

 Reviewer #2: 

a I recommend a critical read to correct grammatical errors/omissions/repetitions/ and omissions. e.g.

a. …was significantly higher in the Middle and Coastal zones (p<0.001), among older adolescents (p<0.001). 

 “was” has been changed to “were”, and an “and” conjunction introduced before among…

 38

b The design and methods used makes it… 

 This has been corrected to read “The design and methods used to make it …” 92

c . … that affect sexual…

 that aff ect sexual… has been corrected, and now reads “that affect sexual…”

 69

d. …first with the the sorted… The repetition has been corrected and now reads “first with the sorted strata generated from administrative regions ….”

 98

e . …with the the sorted… changed to “the first step ensures a sorted…” The repetition has been corrected and now reads “first with the sorted strata generated from administrative regions ….” 98

f. kenya. kenya has been corrected to Kenya

 236

 ABSTRACT

2 Design: The authors should clarify the sample for the study, thus what age frame was the caveat? 

 From a total of 25 062 women respondents, 20 277 (80.9%) women were aged 20 years and above whiles 4785 (19.1%) were adolescents aged between 15-19 years. Thus data from these 4785 adolescents aged between 15-19 years were used for this secondary data analysis.

 26-31

3 Motherhood as conceptualised for the study should be defined in this section. 

 Both adolescent pregnancy and motherhood have been defined in this section. It reads “Adolescent pregnancy was defined as adolescents who have ever been pregnant, whiles adolescent motherhood was defined as adolescents who have ever given birth”

 28-30

4 Results: In the first statement, i.e. “Of the 25062 women included in the survey, 4785 were adolescents between 15-19 years.” Does this mean that the study included participants outside adolescence, considering the word “women”. This need to be clarified and if only females in adolescence, then authors should change “women” to “adolescents”. Also, the percentage of 4785 should be captured in the statement.

 The maternal health survey includes all women of reproductive age (i.e. 15-49 years).

In the main survey, a total of 25062 women were enrolled, of which 20277 (80.9%) women were aged 20 years and above whiles 4785 (19.1%) were adolescents aged between 15-19 years. Only adolescents aged 15-19 years were included in this secondary data analysis for the purpose of our research objectives. This has been clarified in the design section. We have also included a flow diagram that highlights the respondents of this secondary analysis. 32-33,104-108

5 Conclusion: Write these in full “AP and AM.” 

 AP and AM have been written in full as adolescent pregnancy and adolescent motherhood.

 43-47, 301-306

6 From the focus of the study and your findings, it is expected that you make specific recommendations that address your “socio-economic” findings, however, not even a single recommendation was made in this regard. The recommendations provided must therefore be reconsidered. The recommendations have been revised to reflect the objectives and results of our study.

 45-47, 301-306

 INTRODUCTION

7 Page 9: Change from “The Social sequelae…” to “The social sequelae…” 

 This has been amended

 69

8 Page 10: Remove the misplaced “s” from “…births are s still on the rise…” 

 The misplaced “s” has been removed

 75

9 Page 10: Authors should reconstruct this statement for clarity “It is important to note that relatively little research has been conducted on the socio-economic consequences of adolescent mothers and their children in Ghana.” This sentence has been clarified. It now reads “Although there are several studies on adolescent pregnancy in Ghana, few studies have examined the association of socio-economic factors on adolescent pregnancy and motherhood using data from nationally representative surveys.”

 81-84

 METHODS

10 GMHS” should be written in full at first use. Ghana Maternal Health Survey (GMHS) has been written in full at first use

 90-91

11 The last paragraph under the “METHODS” section should be removed and be placed under a subheading “Ethical Consideration”, possibly right after the statistical analysis. 

 The last paragraph of the methods section has been removed and placed under the subheading “Ethical Consideration”

 143-147

 RESULTS 

12 1. Change from “RESULT” to “RESULTS” This change has been effected

 149

13 Instead of “adolescents”, some sections of the results mention “women” and this must be revised. Sections with “women” changed to “adolescents”

 Entire manuscript

14 Authors should report the standard deviation in addition to the mean age. 

 The standard deviation has been reported together with the mean age. The sentence now reads “The mean age of the 4758 respondents included in this analysis was 17 (Standard Deviation (SD) =1.40) years”. 152-153

15 Characteristics of study participants by pregnancy and birth history

“…with the highest prevalence in the Brong Ahafo Region and the lowest

prevalence in the Greater Accra region.” Please quote the percentages and CIs. 

 The percentages and CIs have been quoted. The sentence now reads “… with the highest prevalence in the Brong Ahafo Region (18.6% CI:14.9% - 22.4%) and the lowest prevalence in the Greater Accra region (8.1% CI: 6.1% - 10.2%)”

 163-169

16 Association of socio-economic factors with adolescent pregnancy (AP)/ Association of socio-economic factors with adolescent motherhood (AM)

Authors reported some adjusted findings in these sections, yet the odds ratios are quoted as though they are results from unadjusted models. This should be rectified for clarity.

 The results from the adjusted odds ratios have been clarified. 

 188-198

 CONCLUSION

 The conclusions are great; however, the recommendations do not align with findings of the study and the prime focus of the study. Recommendations are to be made regarding the socio-economic factors studied. The recommendation modified to read as “Our analysis revealed that education, age, household wealth, marital status and zone of residence are associated with adolescent pregnancy and adolescent motherhood in Ghana. Given that adolescent pregnancy and motherhood were significantly higher in the Middle and Coastal zones, and among older adolescents, sexual and reproductive health education should be intensified among these populations. Also, adolescent-friendly corners should be made available and accessible to all adolescents in Ghana irrespective of where they live or their age”. 

 301-306

---

## [Editor Report · Decision Letter 1]

13 Jul 2022

Socio-economic factors associated with adolescent pregnancy and motherhood: analysis of the 2017 Ghana maternal health survey

PONE-D-21-40893R1

Dear Dr. Senkyire,

We’re pleased to inform you that your manuscript has been judged scientifically suitable for publication and will be formally accepted for publication once it meets all outstanding technical requirements.

Kind regards,

Nguyen Toan Tran

Academic Editor

PLOS ONE

Additional Editor Comments (optional):

There are still a few minor English typos (e.g., "adolescet"). Please ensure a thorough read and correction of typos in the final typeset version.
---

## [Editor Report · Acceptance letter]

2 Aug 2022

PONE-D-21-40893R1 

Socio-economic factors associated with adolescent pregnancy and motherhood: analysis of the 2017 Ghana maternal health survey 

Dear Dr. Senkyire:

I'm pleased to inform you that your manuscript has been deemed suitable for publication in PLOS ONE. Congratulations! Your manuscript is now with our production department. 

Kind regards, 

on behalf of

Professor Nguyen Toan Tran 

Academic Editor

PLOS ONE